Phylogenetic relationships, origin and historical biogeography of the genus Sprattus (Clupeiformes: Clupeidae)

Canales-Aguirre Cristian B. cristian.canales@ulagos.cl 1 2
Ritchie Peter A. 3
Hernández Sebastián 4 5
Herrera-Yañez Victoria 6
Ferrada Fuentes Sandra 6
Oyarzún Fernanda X. 1 7 8
Hernández Cristián E. 9 10
Galleguillos Ricardo 6
Arratia Gloria 11
1 Centro i∼mar, Universidad de Los Lagos , Puerto Montt , Chile
2 Núcleo Milenio INVASAL , Concepción , Chile
3 School of Biological Sciences, Victoria University of Wellington , Wellington , New Zealand
4 Biomolecular Laboratory, Center for International Programs, Universidad Veritas , San José , Costa Rica
5 Sala de Colecciones Biológicas, Facultad de Ciencias del Mar, Universidad Católica del Norte , Coquimbo , Chile
6 Laboratorio de Genética y Acuicultura, Departamento de Oceanografía, Facultad de Ciencias Naturales y Oceanográficas, Universidad de Concepción , Concepción , Chile
7 Centro de Investigación en Biodiversidad y Ambientes Sustentables (CIBAS), Universidad Católica de la Santísima Concepción , Concepción , Chile
8 Instituto Milenio en Socioecología Costera , Santiago , Chile
9 Laboratorio de Ecología Evolutiva y Filoinformática, Departamento de Zoología, Facultad de Ciencias Naturales y Oceanográficas, Universidad de Concepción , Concepción , Chile
10 Universidad Católica de Santa María , Arequipa , Perú
11 Biodiversity Institute and Department of Ecology & Evolutionary Biology, University of Kansas , Lawrence , United States of America
Edwards Scott
Electronic publication date: 2021 Aug 18
Publication date: 2021
Volume: 9
Electronic Location ID: e11737
Received 2020 Oct 7; Accepted 2021 Jun 17
Copyright: ©2021 Canales-Aguirre et al.
Copyright year: 2021
Copyright holder: Canales-Aguirre et al.
License: This is an open access article distributed under the terms of the Creative Commons Attribution License, which permits unrestricted use, distribution, reproduction and adaptation in any medium and for any purpose provided that it is properly attributed. For attribution, the original author(s), title, publication source (PeerJ) and either DOI or URL of the article must be cited.
License URL: https://creativecommons.org/licenses/by/4.0/

Keywords: Antitropical distribution, Sprat, Molecular clock, Clupea, BEAST

Funding: FIPA 2010-17 CONICYT doctoral Internship, the FONDECYT 11180897 1201506 ANID –Millennium Science Initiative NCN16_034 This work was supported by the FIPA 2010-17, the CONICYT doctoral Internship, the FONDECYT 11180897, 1201506 and by ANID –Millennium Science Initiative –NCN16_034. The funders had no role in study design, data collection and analysis, decision to publish, or preparation of the manuscript.

==============================
The genus Sprattus comprises five species of marine pelagic fishes distributed worldwide in antitropical, temperate waters. Their distribution suggests an ancient origin during a cold period of the earth’s history. In this study, we evaluated this hypothesis and corroborated the non-monophyly of the genus Sprattus, using a phylogenetic approach based on DNA sequences of five mitochondrial genome regions. Sprattus sprattus is more closely related to members of the genus Clupea than to other Sprattus species. We also investigated the historical biogeography of the genus, with the phylogenetic tree showing two well-supported clades corresponding to the species distribution in each hemisphere. Time-calibrated phylogenetic analyses showed that an ancient divergence between Northern and Southern Hemispheres occurred at 55.8 MYBP, followed by a diversification in the Oligocene epoch in the Northern Hemisphere clade (33.8 MYBP) and a more recent diversification in the Southern Hemisphere clade (34.2 MYBP). Historical biogeography analyses indicated that the most recent common ancestor (MRCA) likely inhabited the Atlantic Ocean in the Southern Hemisphere. These results suggest that the ancestral population of the MRCA diverged in two populations, one was dispersed to the Northern Hemisphere and the other across the Southern Hemisphere. Given that the Eocene was the warmest epoch since the Paleogene, the ancestral populations would have crossed the tropics through deeper cooler waters, as proposed by the isothermal submergence hypothesis. The non-monophyly confirmed for the genus Sprattus indicates that its systematics should be re-evaluated.

Introduction

Antitropical distribution patterns—when closely related taxa have geographic distributions to the north and south of the tropics, but not within—are an active line of research in evolutionary biogeography that can benefit greatly from using congeneric species in phylogenetic context. Congeneric species share a common history from their ancestral population, and several studies have shown that the combined analyses of biogeographic history and time-calibrated phylogenies in congeneric species provide a greater insight into the evolutionary processes involved (e.g., Lavoué et al., 2013). There are still important ecological and commercial fish genera with antitropical distribution patterns that remain to be studied, such as the genus Sprattus.

The five extant species currently assigned to the genus Sprattus (Fig. 1; Clupeiformes, Clupeidae, Clupeinae) are small marine pelagic fishes that inhabit coastal areas and are well known for their schooling behavior (Whitehead, 1988; Fricke, Eschmeyer & Van der Laan, 2021). They are important components of several food webs and some species are commercially important (Frederiksen et al., 2006). These species occur in cooler waters and have an antitropical distribution (Whitehead, 1988; Fig. 1). Sprattus sprattus (Linnaeus, 1758) is the most widely distributed species, and it is the only species in the genus found in the Northern Hemisphere, mainly around the coasts of Europe (Whitehead, 1988; Fricke, Eschmeyer & Van der Laan, 2021). Sprattus fuegensis (Jenyns, 1842) is found on the South American coast, mainly in the Patagonian shelf from the Pacific and Atlantic Oceans (Whitehead, 1988; Aranis et al., 2007; Canales-Aguirre et al., 2016; Fricke, Eschmeyer & Van der Laan, 2021). The other three species are found in Oceania: S. novaehollandiae (Valenciennes, 1847) in south-eastern Australia, and S. antipodum (Hector, 1872) and S. muelleri (Klunzinger, 1879) on the coast of New Zealand (Whitehead, Smith & Robertson, 1985; Whitehead, 1988; Fricke, Eschmeyer & Van der Laan, 2021).

Phylogenetic analyses have shown that the genus Sprattus is sister to the genus Clupea (Lavoué et al., 2007; Li & Ortí, 2007), and it has been suggested that they diversified between 2.66–6.75 MYBP (Jérôme et al., 2003; Cheng & Lu, 2006), which is consistent with the Miocene record of Clupea. Moreover, the extant Clupea species are thought to have radiated during the Pliocene (3.3–3.5 MYBP; Grant, 1986; Wilson, Teugels & Meyer, 2008), which is when the genus Sprattus is thought to have diverged. More recent studies based on large fossil-calibrated phylogenies suggested that the genus Sprattus is a paraphyletic group, and S. sprattus is more closely related to Clupea spp. than to its relatives in the Southern Hemisphere (Lavoué et al., 2013; Bloom & Lovejoy, 2014; Egan et al., 2018).

Figure 1 Distributional map of extant Sprattus and closely related species used in this study.

Red dashed line represents S prattus fuegensis; brown is S. novaehollandiae; orange is S. muelleri; light blue is S. antipodum; and green is S. sprattus. Yellow solid line represents Clupea harengus; purple is C. pallasii; gray is Ramnogaster melanostoma; and blue is Strangomera bentincki.

No study has examined the biogeographic origin of the genus Sprattus; though, information of species with similar antitropical distribution pattern have been conducted. For example, studies of extant populations of Sardinops species showed a recent diversification event between 0.2–2 MYBP (Grant & Leslie, 1996; Bowen & Grant, 1997; Grant & Bowen, 1998), whereas species included in the genus Engraulis diversified between 5–10 MYBP (Grant, Leslie & Bowen, 2005). When considering marine species that have an antitropical distribution, the tropical zone appears to act as a barrier to long-distance dispersal, restricting gene flow between the Northern and Southern Hemispheres (Grant, Lecomte & Bowen, 2010). Experimental studies aiming to evaluate the thermal tolerance of two temperate species of Clupeidae (e.g., Clupea harengus and Sardinops sagax) evidenced their low tolerance for warm (tropical) waters (Martínez-Porchas, 2009; Peck et al., 2012). These results reinforce the hypothesis that warm waters act as a dispersal barrier.

Considering the current antitropical distribution pattern of the genus Sprattus, we hypothesize that the lower sea temperatures of the tropics during the cooler glacial periods between the Miocene and Pliocene might have provided a window of opportunity for the most recent common ancestor of Sprattus to disperse to the other hemisphere. In this study we test the origin and the monophyly of the genus Sprattus using a phylogenetic approach based on DNA sequences from five mitochondrial genome regions (mtDNA). We also examine the historical biogeography of the group, and we used a molecular clock to determine the pattern and timing of species diversification.

Material and Methods

Taxon sampling

Sprattus species have a least concern status for the IUCN Red List and are not listed under CITES. We did not kill fishes for the purpose of this study; instead, tissue samples were provided by researchers worldwide. Unfortunately, samples for Sprattus novaehollandiae were impossible to obtain, therefore we used only three Sprattus species from the Southern Hemisphere. All tissue samples arrived fixed in ethanol 90%, and their general capture locations were S. fuegensis (n = 7) from Chilean fjords in the Southeast Pacific Ocean, S. sprattus (n = 5) from Norwegian fjords in the Northeast Atlantic Ocean, S. muelleri (n = 4) from Auckland Harbour, and S. antipodum (n = 1) from Wellington Harbour (New Zealand).

DNA extraction, PCR and DNA sequencing

Total genomic DNA was dissolved in a buffer containing proteinase K and SDS detergent, and then extracted using a standard phenol-chloroform protocol (Sambrook et al., 1989). DNA was precipitated in 70% ethanol and resuspended in 50 µL of TE buffer. DNA was quantified using a NanoDrop ND-1000 spectrophotometer and diluted to a concentration of 20 ng/µL.

Five mitochondrial fragments were amplified using genus-specific primers (762 bp for Cytochrome b, CytB; 857 bp for Cytochrome Oxidase subunit I, COI; 827 bp for NADH dehydrogenase subunit 2, ND2; and 348 bp for NADH dehydrogenase subunit 3, ND3) designed in this study, and one primer pair described previously (1107 bp for Control Region, CR; Palumbi et al., 1991; Bernatchez, Guyomard & Bonhomme, 1992; see Supporting Information Table S1). The genus-specific primers were designed from the complete mitochondrial genomes sequences deposited in GenBank: Sprattus sprattus (NC009593), S. muelleri (NC016669) and S. antipodum (NC016673). For CytB, COI, ND2, and ND3 fragments, the PCRs were conducted in 30 µL volumes containing 1X PCR Buffer (Invitrogen®; Tris–HCl 200 mM, pH 8,4, KCl, 500 mM), 3 mM MgCl2, 0.2 mM of each dNTP’s, 0.2 µM of each primer, 0.4 mg/mL of BSA, 1.5 units of Taq DNA polymerase (Invitrogen®), and 2 ng of genomic DNA. Thermal cycling was performed in an MJ Research PTC-200 Thermal Cycler with the following parameters: 95 °C for 180 s, followed by 35 cycles of 94 °C for 30 s, 55 °C for 30 s, 74 °C for 60 s, and a final extension at 74 °C for 300 s. For CR the PCR was amplified using 2 mM MgCl2 and the thermo cycling parameters: 94 °C for 300 s, followed by 35 cycles of 94 °C for 30 s, 54 °C for 60 s, 74 °C for 90 s, and a final extension at 74 °C for 600 s. PCR products were purified with ExoSAP-IT® following manufacturer’s guidelines and sequenced in both directions using an ABI 3730xl Genetic Analyzer (Massey University Genome Sequencing Service). Sequences were deposited in GenBank database under the accession numbers MW075156- MW075219. Additional sequences to genus Sprattus were included in the ingroup for further analyses: (i) Clupea harengus (KC193777) and Clupea pallasii (AP009134), including two herring subspecies from C. pallasii (C. p. marisalbi and C. p. suworowi), given their close relatedness to the genus Sprattus; (ii) Ethmidium maculatum (AP011602), Ramnogaster melanostoma (GQ890211- GQ890214, KU288994 –KU288995), and Strangomera bentincki (MW075156-MW075219), given their close relatedness with the Sprattus-Clupea clade; (iii) Potamalosa richmondia (AP011594) and Hyperlophus vittatus (AP011593), because they are more distantly related genera to the Sprattus-Clupea clade; and (iv) Sprattus sprattus (AP009234), Sprattus muelleri (AP011607), and Sprattus antipodum (AP011608) to increase the number of sequences of our target genus. As outgroups, we included Gilchristella aestuaria (AP011606) and Ehirava fluviatilis (AP011588), two species of the subfamily Ehiravinae used for rooting and time calibration purposes.

Initial alignment was performed in Geneious® 6.0.5 (Kearse et al., 2012), and the final alignment was adjusted by eye. Phylogenetic analyses were conducted separately on each gene (to compare each gene tree) and concatenated fragments (because mitochondrial DNA constitutes a single heritable unit). Divergence time and historical biogeography analyses were conducted using a concatenated alignment of the five mitochondrial fragments. Our concatenated data matrix included 13 sequences (one taxa each) and 3,228 characters.

Phylogenetic analyses and divergence time

Before conducting the phylogenetic analyses, we performed Xia’s test implemented in DAMBE v5 (Xia et al., 2003; Xia, 2013) to evaluate whether the DNA sequences we used showed evidence of saturation by substitution (i.e., back mutations), which would need to be corrected using a model of sequence evolution during the phylogenetic analyses. We estimated and compared a substitution saturation index with a critical substitution saturation index (Xia et al., 2003; Xia, 2013) to test that the data set is informative for performing phylogenetic analyses. The results of Xia’s test suggest that there is a low level of saturation in our data set, where the critical index of substitution saturation values was significantly higher than the observed index of substitution saturation values (Supporting Information Table S2).

We ran a Bayesian Markov Chain Monte Carlo (BMCMC) phylogenetic analysis that included a general likelihood-based mixture model of gene-sequence evolution and a Reversible-Jump Markov Chain Monte Carlo procedure (Pagel & Meade, 2004; Pagel & Meade, 2006; Pagel & Meade, 2008; Gascuel, 2005). This phylogenetic reconstruction was implemented in BayesPhylogenies v1.1 software (Pagel & Meade, 2004). This approach enables possible models and parameters to be explored, converging towards the model that best fits the data in the sample of posterior trees (Pagel & Meade, 2008). We ran five independent chains using 106 generations, sampling every 10,000th tree sample, and burning the first 25% of the trees. Finally, we obtained the phylogenetic consensus tree using 750 tree samples.

Approximate divergence times among Sprattus species were estimated using a Bayesian approach implemented in the BEAST v2 software (Heled & Drummond, 2008; Drummond et al., 2012; Bouckaert et al., 2014). To obtain divergence times, we used the Log-Normal Relaxed Clock Model (LNCM; Drummond et al., 2006; Drummond & Suchard, 2010). We ran this model five times using the most complex sequence evolution model, GTR+I+G, with 10,000,000 generations sampling each 10,000 generations. The outputs of each run were combined in LogCombiner software to increase the Effective Sample Size (ESS) to be at least >200. The ESS of a parameter sampled from an MCMC is the number of effectively independent draws from the posterior distribution of the Markov Chain.

To obtain the posterior distribution of the estimated divergence time, the age of a fossil, †Lecceclupea ehiravaensis, dated during the late Campanian in the Late Cretaceous epoch at about 74 MYBP was used. (Taverne, 2011 interpreted this age as part of the Campanian-Maastrichtian; however, 74 MYBP is currently considered within the Campanian according to the ICS International Chronostratigraphic Chart, 2021; http://www.stratigraphy.org.) This age was used as a calibration point to constrain the age in the Gilchristella aestuaria and Ehirava fluviatilis node. †Lecceclupea ehiravaensis has been shown to be a crown member of the clade (Ehirava, Gilchristella; see Taverne, 2011). Prior age distribution of this clade follows a lognormal distribution using the age boundaries of the geological stage from which the fossil was excavated (i.e., 95% credibility interval). An offset of 74 MYBP was applied to the model. Subsequently, we used the Log-Normal Relaxed Clock Model and previous set parameters to run 10 independent Markov Chain Monte Carlo (MCMC) simulations with a chain length of 107 generations. Sampling was conducted every 10,000 generations and we used as prior distributions the following parameters: the base frequency, proportion invariant sites, and proportions of each transition and transversion, all of those to increase the effective sample size. The individual runs were combined using LogCombiner burning 250 trees per each sample. Finally, a maximum clade credibility tree was created in TreeAnnotator, which enable a summary tree to be visualized in FigTree v1.4 (https://github.com/rambaut/figtree/releases).

Historical biogeography

We inferred the historical distribution of the genus Sprattus and its close relatives using their current distribution (i.e., longitude and latitude as continuous traits). This approach was chosen over the multistate discrete data for the following reasons: (i) discrete data could bias the ancestral state of a descendant species distributed in the same geographical region; (ii) continuous data permit identifying dispersal trends; and (iii) classical discrete multistate estimation does not consider the spherical nature of the earth (O’Donovan, Meade & Venditti, 2018; Gardner, Surya & Organ, 2019; Avaria-Llautureo et al., 2021). For these, we used the current geolocation to infer the ancestral distribution for each node of the phylogenetic tree. To reconstruct the distribution, we used the Geo Model (O’Donovan, Meade & Venditti, 2018) and implemented BayesTraits v3.0 (Pagel & Meade, 2004). The Geo Model estimates the posterior distribution of their geo-position across phylogenetic nodes. We used tree samples obtained in BMCMC phylogenetic analyses and a trait matrix. We ran 106 generations sampled every 10,000 generations to obtain a parameters sample. Posteriorly, a 25% burned-in was used to avoid including parameters sampled before the convergence of the Markov Chain, and a final sample of 750 parameters was obtained. The ancestral distribution of each node was plotted on a paleogeographical perspective using mapast v0.1 R package (Varela & Rothkugel, 2018). We combine paleomaps from 10, 30, 50, 90, 110 MYBP using SETON2012 as a global plate motion model (Seton et al., 2012).

Results

Phylogenetic tree reconstructions using the concatenated fragments (Fig. 2) and each mitochondrial fragment independently showed a similar pattern (Fig. S1). Each extant Sprattus species forms a monophyletic group. The Sprattus species were distributed in the phylogenetic tree in two main clades that matched their antitropical distribution, each in one hemisphere. The Northern Hemisphere clade included Sprattus sprattus and the species Clupea harengus and C. pallasii (including their subspecies); the Southern Hemisphere clade included Sprattus fuegensis, S. antipodum, S. muelleri, Ramnogaster melanostoma, and Strangomera bentincki. However, overall, the genus Sprattus is polyphyletic, because S. sprattus is closely related to Clupea and S. fuegensis, whereas S. antipodum and S. muelleri are closely related to Ramnogaster and Strangomera.

Figure 2 Bayesian consensus tree concatenating of mitochondrial genes from 750 more likely trees.

Branch lengths are proportional to the number of substitutions per nucleotide position. Numbers at nodes are posterior probabilities from Bayesian analyses. Grey rectangles indicate current hemisphere distribution. Red branches for S. prattus fuegensis, orange for S. muelleri, light blue for S. antipodum, and green for S. sprattus.

The time-calibrated phylogenetic analyses showed a divergence between Northern and Southern Hemispheres that was dated at 55.8 MYBP (early Eocene; Fig. 3A). There was also another diversification event among the Northern Hemisphere clade at 33.8 MYBP (boundary between Eocene and Oligocene), splitting Sprattus sprattus from Clupea species. Current species of Clupea diverged about 8.5 million years ago (late Miocene). For the Southern Hemisphere clade, species diverged at 33.2 MYBP (early Oligocene). Among the species of the Southern Hemisphere clade, Strangomera bentincki split from other Sprattus species around 22.6 MYBP (early Miocene), Sprattus fuegensis split at 13.3 MYBP (middle Miocene) from their New Zealand relatives, and the most common recent ancestor of S. antipodum and S. muelleri diverged around 5.6 MYBP (boundary between Miocene and Pliocene). Ancestral distributions (Figs. 3B–3G) show that the MRCA of the Northern and Southern clades likely inhabited the Southern Hemisphere in the Atlantic Ocean (Fig. 3D).

Figure 3 Time-calibrated phylogenetic tree based on Bayesian relaxed-clock analyses (A) and reconstruction of the ancestral geopositioned nodes (B –G).

Numbers at nodes are divergence time since the root of each species. Horizontal colored bars indicate the 95% HPD of divergence times, and the scale axis shows divergence times as millions of years ago (MYBP). Analyses based on the topology and branch lengths of the Bayesian phylogenetic trees. Colored dots in B –G correspond to posterior distribution of ancestral locations measured in longitude and latitude. Colors are associated to horizontal-colored bars in (A). Paleomap reconstructions from 10, 30, 50, 90, 110 MYBP were obtained using SETON2012 global plate motion models.

Discussion

Non-monophyletic genus Sprattus

We confirmed that the genus Sprattus is a polyphyletic group with an antitropical distribution, challenging the taxonomic status of the Sprattus species. Considering the two geographic clades in opposing hemispheres, the Northern clade closely relates S. sprattus with the genus Clupea, and the Southern clade closely relates the rest of Sprattus members with Strangomera bentincki and Ramnogaster melanostoma. The relationship among species from the Southern clade has not been described before. This taxonomic incongruence in the genus Sprattus has also been identified in studies that use large phylogenies in Clupeiformes and have focused in the identification of the biogeographic or diadromy origin, body size, dispersal pattern, or trophic niche evolution of the group (Lavoué et al., 2013; Bloom & Lovejoy, 2014; Egan et al., 2018; Bloom, Burns & Schriever, 2018; Avaria-Llautureo et al., 2021). Although some of these studies are based on DNA of different types (i.e., mt or nDNA) or taxa (i.e., Sprattus members and its close relatives), they support the polyphyly of the genus Sprattus. Therefore, our results provide further support for Sprattus being polyphyletic and add S. fuegensis and Strangomera bentincki as pieces of the puzzle to understand the evolution in the Southern clade.

Taxonomic classification and phylogenetic relationships among the genera Sprattus, Ramnogaster, Strangomera, and Clupea are unclear if they are only based on morphological and meristic traits. All these taxa resemble the Clupea type and were first classified as species of Clupea ( Whitehead, Smith & Robertson, 1985; Whitehead, 1988). The genus Sprattus was erected by Girgensohn (1846) based on S. haleciformis, which was later synonymized with S. sprattus (Whitehead, 1988), defining the absence of a pterotic bullae as the key diagnostic feature (Mathews, 1884; Whitehead, 1964; Whitehead, 1988; Whitehead, Smith & Robertson, 1985). However, fewer pelvic rays and an anteriorly placed pelvic fin (Whitehead, 1988) have also been used to differentiate Sprattus from Clupea. The two genera also differentiate in key reproductive traits, whereas Sprattus produces pelagic eggs, Clupea produces demersal eggs that attach to the seabed or vegetation (Haegele & Schweigert, 1985; Whitehead, 1988). Finally, the genera Sprattus and Ramnogaster share the absence of pterotic bullae, but differ in fin-ray numbers, whereas Sprattus differs from Strangomera on having more gill rakers (Whitehead, 1988).

Incomplete sorting lineage, introgression, or convergence of morphological traits could be plausible explanations for the current Sprattus taxonomic classification and our gene tree. The first two can be ruled out, because none of the species of this study shared or had similar haplotypes. Introgression may also be ruled out, because the fishes have different reproductive strategies: pelagic or demersal eggs (Haegele & Schweigert, 1985; Whitehead, 1988), so there is little opportunity for cross-fertilization. However, introgression could be true if divergence in reproductive ecology occurred at an initial stage older than 33.8 MYBP between S. sprattus and Clupea species. Introgression and ancient hybridization events could be identified by comparing mtDNA and nDNA (Saitoh et al., 2011), however, this has not been detected in clupeid phylogenies (Bloom & Lovejoy, 2014). We cannot discard the convergence of morphological traits explanation given that there are traits that look similar and others that support the separation of Sprattus and Clupea (Mathews, 1884; Whitehead, 1964; Whitehead, 1988; Whitehead, Smith & Robertson, 1985).

For Sprattus species from the Southern Hemisphere, we found that S. fuegensis from South America is the sister to New Zealand’s sympatric S. antipodum and S. muelleri. Nonetheless, we need to keep in mind that we could not include S. novaehollandiae, hence further studies should include this species. For New Zealand sprats, it only has been suggested that these species might have different ecological requirements considering their sympatry (Whitehead, Smith & Robertson, 1985). We suggest that further investigations be done to disentangle the mechanisms that promoted sympatric speciation for S. antipodum and S. muelleri.

Divergence time and historical biogeography

The results based on a fossil calibration showed that the two antitropical clades diverged in the Eocene (55.8 MYBP; older than we hypothesized), with a likely origin in the Atlantic Ocean in the Southern Hemisphere. The species within the Northern Hemisphere clade diverged at 33.8 and in the Southern Hemisphere at 33.4 MYBP, during the early Oligocene. Cheng & Lu (2006) and Jérôme et al. (2003) estimated that the divergence event of the two genera occurred between 6.75–2.66 MYBP (late Neogene-early Quaternary). This estimation disagrees with the older divergence time found in our study, which could be explained by the calibrating method used by the authors. Different calibrating methods typically yield different results, and each method has its own particular challenges. In previous studies the authors used a standard nucleotide substitution rate for fish, which is a method that depends on the timescale over which those rates are measured (Hipsley & Müller, 2014) and could generate an overestimation of divergence times (Phillips, 2009; Ho et al., 2011; Hipsley & Müller, 2014). Fossil calibrations do not produce this problem, although the uncertainty in age and phylogenetic position present a different challenge (Hipsley & Müller, 2014). To address this and avoid the overestimation of the divergence time, we ran our analysis based on the fossilized birth-death process calibration method and a Bayesian framework, which included the uncertainty of dating species divergences and yield with more accurate node age estimates (Heath, Huelsenbeck & Stadler, 2014; Bouckaert et al., 2014; Gavryushkina et al., 2017).

The Eocene was the warmest geological epoch of the last 65 million years (Zachos et al., 2001), where sea surface temperatures in the Atlantic tropical areas may have been up to 38 °C (Cramwinckel et al., 2018). The ancestor of Clupeoidei originated and diversified in the tropical Indo-West Pacific region during the Lower Cretaceous (119 MYBP, Lavoué et al., 2013), and it would have been adapted to warm, marine temperatures (i.e., >25 °C; Lavoué et al., 2013; Bloom & Lovejoy, 2014). Considering this, our analyses show that the Clupeidae lineage spread to the Southern Hemisphere earlier than the clades that included Sprattus, Clupea and close relatives. Similarly, the species of Potamalosa, Hyperlophus and Ethmidium also inhabit temperate waters in the Southern Hemisphere, suggesting that this old south-distributed group of fishes was able to cross the tropics but not to adapt to the warmer environment. Nonetheless, extant members of the genera Sprattus and Clupea are now distributed antitropically in much colder temperate waters (Whitehead, Smith & Robertson, 1985; Lavoué et al., 2013), and although they mainly inhabit marine environments (Bloom & Lovejoy, 2014), they can also inhabit areas with highly variable environments, such as fjords (e.g., Glover et al., 2011; Canales-Aguirre et al., 2016; Canales-Aguirre et al., 2018).

Antitropical distribution patterns are traditionally explained by dispersal and vicariance mechanisms (Stepien & Rosenblatt, 1996; Grant & Bowen, 1998; Burridge, 2002; Le Port, Pawley & Lavery, 2013). Dispersalists have proposed several hypotheses to explain dispersal across the tropics: island integration (Rotondo et al., 1981), dispersal at shallow depths during glaciations (Lindberg, 1991), and isothermal submergence (Hubbs, 1952). Island integration refers to the formation of endemic biotas through the movement of individuals using islands or seamounts (Rotondo et al., 1981). In our case, we can discard this explanation, because clupeids are typically marine and inhabit productive coastal areas (Whitehead, 1988). Dispersal at shallow depths during glaciations is a well-recognized dispersal mechanism for several pelagic fishes during the Pleistocene (Burridge & White, 2000; Burridge, 2002; Grant, Leslie & Bowen, 2005). The isothermal submergence hypothesis refers to the possibility that marine organisms adapted to cool or temperate areas are able to disperse across the tropical region through deeper, colder tropical waters (Hubbs, 1952). Taking into account that the MRCA of these two clades diversified in the warm Eocene, and then each clade diversified between the late Eocene and early Oligocene epochs the isothermal submergence hypothesis seems to be the most plausible explanation. This later because the temperatures begin to decrease until initiation of Antarctic glaciation (Zachos et al., 2001) and some clupeoids, such as herrings, may dive as much as 200 m (Blaxter, Denton & Gray, 1981). Vicariant mechanisms such as plate tectonic, relictual distribution, and equatorial isolation by climatic change or biological interactions have been advocated by others studies (Stepien & Rosenblatt, 1996; Saitoh et al., 2011). However, mechanisms associated with plate tectonics are not supported by our results, because the divergence time among nominal species of Sprattus and Clupea would have occurred during the Eocene, and the present continental configuration closely resembles the configuration of the continents during that time. Studies, such as those by Grant & Bowen (1998) and Grant, Leslie & Bowen (2005) on marine pelagic fishes have supported a dispersalist mechanism to explain the antitropical distribution and exclude vicariant explanations as well.

Dispersion from their ancestral habitat involved adaptation to colder waters, while simultaneously expanding their tolerance to fluctuations in salinity, allowing them to also colonize low saline habitats. The warmer equatorial waters have remained as a key barrier to dispersal between hemispheres, which has only been crossed when windows of colder environments appeared across the tropics or, more plausibly, by using deeper, colder tropical waters as proposed by the isothermal submergence hypothesis.

Supplemental Information

Supplemental Information 1 Primers used in this study

Click here for additional data file.

Supplemental Information 2 Comparison of the index of substitution saturation (ISS) with the critical index of substitution saturation (ISSc) that defines a threshold for significant saturation in the data for symmetrical and asymmetrical tree topology

Click here for additional data file.

Supplemental Information 3 Bayesian consensus tree of each mitochondrial genes and concatenated data

Branch lengths are proportional to the number of substitutions per nucleotide position. Numbers at nodes are posterior probabilities from Bayesian analyses.

Click here for additional data file.

Supplemental Information 4 Sequences used in this study

Click here for additional data file.

The authors are grateful to Jeff Shima from Victoria University of Wellington, Tom Trnski from Auckland War Memorial Museum, and Malcom Francis from National Institute of Water and Atmospheric Research (NIWA) all of them from New Zealand; Kevin Glover from Institute of Marine Research from Norway, and Anna Semenova from Moscow State University from Russia, for sharing tissue samples of the genera Sprattus or Clupea. We thank Sébastien Lavoué and one anonymous reviewer for providing valuable comments on earlier versions of the manuscript and Terry J. Meehan (Lawrence, Kansas, USA) for checking and improving the language or the English style.

Additional Information and Declarations

Competing Interests

Author Contributions

Animal Ethics

Data Availability

The authors declare there are no competing interests.

Cristian B. Canales-Aguirre conceived and designed the experiments, performed the experiments, analyzed the data, prepared figures and/or tables, authored or reviewed drafts of the paper, and approved the final draft.

Peter A. Ritchie, Cristián E. Hernández and Ricardo Galleguillos conceived and designed the experiments, authored or reviewed drafts of the paper, and approved the final draft.

Sebastián Hernández performed the experiments, analyzed the data, authored or reviewed drafts of the paper, and approved the final draft.

Victoria Herrera-Yañez and Sandra Ferrada-Fuentes performed the experiments, analyzed the data, prepared figures and/or tables, and approved the final draft.

Fernanda X. Oyarzún and Gloria Arratia analyzed the data, prepared figures and/or tables, authored or reviewed drafts of the paper, and approved the final draft.

The following information was supplied relating to ethical approvals (i.e., approving body and any reference numbers):

This research was not approved by any Institutional Animal Care because tissue or fin-clipped were provided from personal collections kept by researchers worldwide. Thus, we did not kill fishes for the purpose of this study.

The following information was supplied regarding data availability:

The sequences are available at NCBI: MW075156–MW075219, and MW075156–MW075219, and the raw measurements are available in the Supplementary File.

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
