# Peer review of "Phylogenetic relationships, origin and historical biogeography of the genus Sprattus (Clupeiformes: Clupeidae)"

_PeerJ, doi:10.7717/peerj.11737_

## Round 0.1 · original submission · Major Revisions

The two reviewers agree that this is potentially an interesting paper. However, reviewer 1 suggests broadening the introduction, in addition to other minor issues. Reviewer 2 suggest broadening the taxon sampling using freely available data online. Although this would require some additional analysis it is warranted, I think. It should not be too burdensome to re-do some of the analyses to make a stronger paper.

Reviewer 1 ·

Basic reporting

Letter to Authors
peerj-53259-v0
Phylogenetic relationships, origin and historical biogeography of the genus Sprattus (Clupeiformes: Clupeidae)
Cristian Canales, Peter A Ritchie, Sebastian Hernandez, Victoria Herrera, Sandra V Ferrada Fuentes, Fernanda Oyarzun, Cristian E Hernandez, Gloria Arratia, Ricardo Galleguillos


201018


Dear authors,
This may be a potentially interesting MS dealing with sprat phylogeny and historical biogeography. It is goo to publish in the journal you are submitting to, if adequately written. The major concerns that I have are incomplete taxon sampling, choice of outgroups and interpretation of your results regarding character state reconstruction. Absence of S. novaehollanidae and Ramnogaster from your analysis enables only limited extent of discussion. Too few outgroups needs reanalysis adding some more outgroups. Basal character states seem near fifty-fifty for opposite hypotheses.
This MS has some other minor issues in its writing style and story flow.
Introduction and discussion sections are full of simple reference lists (A stated this, B analyzed that, C argued it, or alike). Exact numerics, species names, localities and time periods in the references do not matter. Use noun phrases to make abstract contents of those references. Abstracting contents of references will understate those of the citing literature and emphasize your current research. I indicated a couple of examples.
See below for detail.

Experimental design

Absence of S. novaehollanidae and Ramnogaster from your analysis enables only limited extent of discussion. Too few outgroups needs reanalysis adding some more outgroups.

Validity of the findings

Basal character states seem near fifty-fifty for opposite hypotheses.

Additional comments

L55 introduction
I am sorry this MS is not interesting. Beginning with your material fish is good for kinky Sprattus geeks. You are submitting to an international multidisciplinary journal. Consider beginning with general statements to draw wider readership. Introduction section also has a function to show a placement of your specific research in a general and wider research field either of marine biology, ichthyology, evolutionary biology or historical biogeography, etc.
An example of a placement of your research may be as follows.
{evolutionary biology of marine organisms,
\t {adaptation to changing environments,
\t \t {evolutionary history of bipolar distribution,
\t \t \t {Sprattus historical biogeography # you are here
\t \t \t }
\t \t }
\t }
};
Make a brief critical review (may be less than 100 words) of the matters at nesting level-1 to 3. Sprattus specific matters will then follow. What research fields would come up to the outer envelopes depends on your choice and insights. You may insert a brief review of recent development of methodology.
Adding a general statements will bloat the text, and you need to make compact the rest of your statements in this section. You may safely reduce detailed but tedious taxonomical review of Splattus over L59-73. You have a figure showing global distribution patterns of Sprattus and relatives, and then you may omit detailed description of their distribution in the text.

L75,76,282,296,299
Sprattus genus -> genus Sprattus
Clupea genus-> genus Clupea

L144
NC009593 -> AP009234
NC entries are of re-annotation of its original annotated sequences. See COMMENT field of the entries. Respect the original.
NC16669 ?
NC16673 ?

L147
20 ng/uL of genomic DNA ?
It is impossible to set up this reaction cocktail unless you added all reagents in powder. See L137.

L156
Clupea pallasi .. were used as outgroup
To stabilize outgroup rooting, more OTUs are necessary. When outgroups are too few, ingroup risks (your case) or misrooting risks (currently unclear) will occur. See L243.

L157
NC009578 (NC_009578 ?) -> AP009134
NC016710 -> ...
NC entries are of re-annotation by NCBI curators from the original. Respect the original. See COMMENT field of the entry.

L161,162
NC -> ...

L259
likely inhabited the North Hemisphere ?
Fig.4A tells it is near neutral.

L269
Fig.4B tells it is near neutral.

L264-273
Delete this simple repetition of your results.

L276
which, challenges -> {which challenges (delete a comma), challenging}

L278
differentiate -> differentiated

L283
however, -> but
only included -> included only

L288
A recent study, that included nuclear loci and a total evidence analyses, -> A recent study based on combined nuclear and mitochondrial loci (delete commas)

L289
Ramnogaster genus -> genus Ramnogaster

L291
although the authors incorporated four genes (two nuclear, Rag1 and Rag2 and two mtDNA, 16S and CytB), (not essential) -> delete
Taxon sampling is essential here.

L293-294
from some of .. and CytB) (not essential) -> delete

L296
confirm -> confirming
previous -> previous hypothesis
not a monophyletic group, but it is paraphyletic -> not [a] monophyletic but a paraphyletic group

L304
, in addition to the pterotic bullae, (redundant) -> delete
This insertion disturbs readers' short-term memory, because the topic of this sentence is not the pterotic bullae.

L305-308
For instance, .. in the genus Clupea (Whitehead, 1985).
Do not make a simple reference list. Use noun phrases to make abstract contents of those references. Abstracting contents of references will emphasize your results. Hint: .. have been used to differentiate the genera Sprattus from Clupea {with, such as} fewer pelvic rays (Whitehead et al., 1985) and anteriorly placed pelvic fin (Whitehead 1985). (Whether six or seven does not matter. Dorsal fin does not matter.)
Please note that this is merely an example. Your introduction and discussion sections are full of simpel reference lists. Make all of them abstract to put your results and arguments forward.

L317
had shared or similar haplotypes -> shared or had similar haplotypes

L318-322
should also be ruled out because ..
This is true only if the divergence in reproductive ecology occurred at an initial stage, older than 16.4 MYA, of speciation between the sprat and herring. As a graduallist, I do not agree with you. The ecological divergence has progressively occurred since 17.6 MYA until 2.0 or 1.5 MYA. Interspecific hybridization was substantiated with current striking ecological differences (Saitoh et al 2010, Kitagawa et al 2011, etc). These studies implicitly tell ecological similarity in old times. Adding brief statements of some reservation is necessary, but do not make it long. Comparison of nuclear sequences is necessary to test if tree topology conflict appear, as in L331-334.

L318
strategies, pelagic -> strategies: pelagic (put a colon)

L336-341
This paragraph tells a different topic and may be in an independent sub-section. You should mention about some reservations under the current analysis without S. novaehollandiae and Ramnogaster.

L349
2.66 - 6.75 MYBP -> 2.66 - 6.75 MYBP with a standard nucleotide substitution rate in fish

L350-352
The molecular dating .. for fish species, (simple reference list) -> delete

L352-353
whereas .. relevant to the two genera. (redundant) -> delete

L361-366
For example, .. 5 - 10 MYBP (Grant and Bowen, 1998; Grant et al., 2005). (simple reference list) -> delete (already abstracted in the previous sentence)

L369-372
Matters in parentheses are of simple reference list. Delete.

L390
likely originated in the Northern Hemisphere .. in the Atlantic Ocean
Fig.4 does not tell that. See my comments on L259,260.

L404
dispersalist and vicariant -> {dispersalist and vicariographer, dispersal and vicariance}

L423
during the cooler Miocene period ?
Miocene period was generally warmer (Herold et al 2012) and the major oceanic cooling down occurred in the Messinian period (Holbourn et al 2018).

L434 references
Check the reference list carefully again from the beginning. Reference lists are frequently den of errors. You might add, omit or swap citation in the main text on the way internal revision. Some might slipped off from the list, or some others might held over the list. It is the authors' responsibility that all references are properly cited.

L449,452,471,535,547,560,
Make sure if paper titles are in lower case.

L527
See L461.

Following items may helpful for further discussions.

Herold N, Huber M, Muller RD, Seton M. 2012. Modeling the Miocene climatic optimum: Ocean circulation, Paleoceanography 27:PA1209.

Holbourn AE, Kuhnt W, Clemens SC, Kochhann KGD, Johnck J, Lubbers J, Andersen N. 2018. Late Miocene climate cooling and intensification of southeast Asian winter monsoon. Nat Commun 9:1584.

Kitagawa T, Fujii Y, Koizumi N. 2011. Origin of the two major distinct mtDNA clades of the Japanese population of the oriental weather loach Misgurnus anguillicaudatus (Teleostei: Cobitidae). Folia Zool 60:343-349.

Saitoh K, Chen W-J, Mayden RL. 2010. Extensive hybridization and tetrapolyploidy in spined loach fish. Mol Phylogenet Evol 56:1001-1010.

Fig.1
How about C. pallasii in the Barents Sea?

·

Basic reporting

No comment.

Experimental design

No comments

Validity of the findings

The dataset should include more taxa to make the taxonomic and biogeographic results and conclusions more inclusive.
Note that all additional data that I suggest to combine with the current authors' data, are freely available (in GenBank). I "only" suggest to redo the same set of analyses that the authors did, but using an expanded dataset, as described below in the "General comments for the authors"

Additional comments

In this study, the authors examined the molecular systematics and biogeography of Sprattus and Clupea, two temperate genera of sardines, using several molecular markers. They found the genus Sprattus paraphyletic relative to Clupea, with the north hemisphere-distributed species, Sprattus Sprattus, more closely related to the genus Clupea than to other south hemisphere-distributed species of Sprattus. The authors, then, discussed the biogeography of the clade (Sprattus, Clupea) and concluded that the most recent common ancestor of this clade has a north hemisphere origin and the ancestors of the lineage (S. muelleri, S. fuegensis and S. antipodum) dispersed to the south hemisphere where they subsequently diversified.

This is an interesting paper which brings new information on the evolution and biogeography of this temperate group of sardines. To my knowledge, this is the first study dealing with the biogeography of this group of temperate sardines. The manuscript is well structured and well written and I have only few comments, but two of my comments are important:

Main comment 1 (about the origin of the clade [Sprattus, Clupea]): The clade (Sprattus, Clupea) belongs to a larger clade of temperate sardines including Ethmidium (already included in this study), Hyperlophus and Pomatalosa as found in Lavoué et al (2013) (but see a different hypothesis in Egan et al (2018)). The distribution of these three genera is restricted to the South hemisphere. It is very likely that the genus Ramnogaster belong to this clade as well (as shortly discussed by the authors see lines 289-293; see also Lavoué et al 2014 and Egan et al 2018 and Whitehead 1964). Ramnogaster is endemic to the marine southern region of South America. Moreover, it is also likely that the genus Strangomera belongs to this clade (the distribution of Strangomera is restricted to south South America and it occurs in commercial quantity along the coast of Chili according to Whitehead (1964, 1985)).
There are two possible consequences of this: 1) (phylogenetic consequence) the clade (Sprattus, Clupea) and the genus Sprattus are likely each not monophyletic relative to Ramnogaster and Strangomera and 2) (biogeographic consequence) if the south-distributed Ramnogaster and Strangomera are closely related to Sprattus and/or Clupea, and if the dataset includes the more distantly related genera, south hemisphere-distributed, Hyperlophus and Pomatalosa, this may significantly impact the ancestral area reconstruction towards a south hemisphere origin.

Main comment 2 (about the time calibration point used). Overall, the timescale of diversification inferred in this study is congruent with those of Lavoué et al (2013) and Egan et al (2018), as already noted by the authors. However, the calibration point used in this study is based on an extinct species (Clupea testis) which is only known by its otoliths. As discussed by Grande (1985) the phylogenetic positions of clupeiform extinct species that are only known from their otoliths (or scales) are difficult (often impossible) to determine. I could not find positive evidence in the literature to classify Clupea testis into the genus Clupea (but I am not a morphologist). It seems, unfortunately, there is no “good” fossil belonging, or closely related, to the clade (Sprattus, Clupea) that can be safely used to calibrate time of divergence of this group (see Whitehead 1985. Whitehead 1985 is a relatively old reference but it seems there is not so much more on this subject, subsequently published).

To overcome these two potential issues, I would like to suggest to expand the taxonomic sampling as following: Hyperlophus vittatus and Potamalosa richmondia (all gene sequences available in GenBank) and Ramnogaster melanostoma (only partial COI sequences [from specimens collected from the Lower Parana River; see Diaz et al 2016] and partial cytochrome b sequences [from specimens collected nearby the Río de la Plata estuary; see Garcia et al 2011 for identification in their Appendix 1] available in Genbank. The quality of the phylogenetic signal of these two genes is good and it should be enough to reliably infer the phylogenetic position of Ramnogaster melanostoma relative to Clupea and Sprattus). Unfortunately, there is still no sequence of Strangomera bentincki available.
As outgroups, it would be interesting (for rooting and time calibration purpose) to add two species of the subfamily Ehiravinae such as Gilchristella aestuaria and Ehirava fluviatilis (sequences available in Genbank). These two species should be constrained to form a monophyletic group (sister group to the rest of the taxonomic sampling) and the time divergence between these two species should be constrained by the age of the fossil species Lecceclupea ehiravaensis (with a strict minimum age of 74 My). Lecceclupea ehiravaensis has been shown to be a crown member of the clade (Ehirava, Gilchristella) (see Taverne 2011).
The ancestral area reconstruction should include all ingroup taxa.

Reference
Díaz J, Villanova GV, Brancolini F, del Pazo F, Posner VM, Grimberg A, et al. (2016) First DNA Barcode Reference Library for the Identification of South American Freshwater Fish from the Lower Paraná River. PLoS ONE 11(7): e0157419. https://doi.org/10.1371/journal.pone.0157419
García, G., G. Martínez, S. Retta, V. Gutiérrez, J. Vergara and M. de las M. Azpelicueta. 2011. Multidisciplinary identification of clupeiform fishes from the Southwestern Atlantic Ocean. Internat. J. Fish. Aqua. 2: 41–52.
Taverne (2011) Les poissons crétacés de Nardo. 33. Lecceclupea ehiravaensis gen. et sp. nov. (Teleostei, Clupeidae). Boll Mus Civ St Nat Verona, sez di Geologia, Paleontologia e Preistoria 35: 3–17.

I think the origin and biogeography of the genus Sprattus cannot be reliably solved without considering these other taxa (as listed above).


Other comments:

Introduction section, lines 75-76: Change “Sprattus genus” to S. sprattus”

Introduction section, line 77: please specify why it is important to indicate that Clupea is morphologically distinctive within Clupeinae (it seems to me that any genus within Clupeinae must be morphologically distinctive to others?)

Introduction section, line 91: maybe change “the same” to “similar”

Introduction section, line 98-101: Maybe this sentence could be reworded. Something like: “Experimental studies aiming to evaluate the thermal tolerance of two temperate species of Clupeidae (i.e. Clupea harengus and Sardinops sagax) evidenced their low tolerance for warm (tropical) water (Martínez-Porchas et al., 2009; Peck et al.,2012). These results are in favour of/reinforce/support the hypothesis that warm waters act as a dispersal barrier”

Introduction section, lines 103-105: Please add a reference supporting this assertion

Introduction section, line 106-107: the sentence part “…S. Sprattus was the earliest diverged species of the group” needs to be reworded

Material and methods, line 128, Sprattus in italics

Material and methods, line 139: please indicate the size of each molecular marker

Material and methods, line 181: please indicate the size of your phylogenetic matrix (characters*taxa)

Material and methods, lines 210-211: See my main comment 2 above: According to Grande (1985:324) Clupea testis is known only from otoliths. Because of that, Grande (1985) questioned its generic assignment/phylogenetic position.

Material and methods, lines 231-232: It seems that the method of ancestral area reconstruction is a method of ancestral character reconstruction which allows only dispersal events (no vicariant events; meaning that the inference only allows one state at node. For example, A or B but not A+B): This is fine but it should be indicated.

Discussion, line 281: change “S. sprattus is phylogenetically more like a Clupea species” to “S. Sprattus is sister to Clupea”

Discussion, line 296: Please change “Sprattus genus is not a monophyletic group, but it is paraphyletic” to “Sprattus genus is a paraphyletic group”

Discussion, line 313: Change “the valid species tree” to “the current generic classification”

Discussion, lines 380-387: I strongly disagree with this explanation for two reasons: 1) Whitehead (1985) who reviewed most of “Clupea” fossils clearly stated that none of these fossils are positively classifiable into the extant genus Clupea (see his detailed discussion on the fossil record of clupeiforms). At the best, “Clupea” fossils are only assignable to the family Clupeidae. 2) based on the timetree of the authors, it is expected to find north hemisphere fossils as old as 16 my (most of the “Clupea” fossils are younger). However, to support a north hemisphere origin of this group, it must be found older (>16my) Clupea/Sprattus-related fossils in the north hemisphere as evidence that this group was present there before the clade (Clupea, S. Sprattus) diversified.

Sebastien Lavoue

---

## Round 0.2 · Minor Revisions

The two reviewers are agreed that minor revisions, mostly grammatical, are necessary. In addition to incorporating their specific comments, please have the manuscript read by a fluent English speaker before submitting the revision so as to improve its clarity throughout.

Reviewer 1 ·

Basic reporting

no comment

Experimental design

no comment

Validity of the findings

no comment

Additional comments

Letter to Authors
peerj-53259-v1
Phylogenetic relationships, origin and historical biogeography of the genus Sprattus (Clupeiformes: Clupeidae)
Cristian B. Canales-Aguirre, Peter A. Ritchie, Sebastian Hernandez, Victoria Herrera-Yanez, Sandra Ferrada-Fuentes, Fernanda X. Oyarzun, Cristian E. Hernandez, Gloria Arratia, Ricardo Galleguillos


210524


Dear authors,
Revision with some reanalysis has made your MS more interesting. Many issues I pointed out last time were cleared, but some others remained or newly introduced. One more round of minor revision is necessary.
See below for detail.


L60
tropical the area -> tropical area

L65
However, there is still (unclear logical link) -> There is

L66
; such is -> such as [in]

L69
, are the important of (not a complete sentence) -> They are [important] components of (break sentence here)

L143
follows -> following

L157
i.e stochastic effects ??
This insertion does not make sense. You should state here why separately. Or otherwise, I do not think separate analysis is necessary. Mitochondrial genome constitutes a single heritable unit, and thus it is a single locus. Then, the longer-the-better rule applies. Concatenated four coding sequences may be the only and the best in your case.

L158
considering that ??
This insertion does not make sense.

L192
(Taverne, 2011) (Taverne 2011) -> (Taverne, 2011)

L202,203
LogCombiner v1.8 ? TreeAnnotator v1.8 ?
Version information may not necessary unless it is not what is implemented in BEAST v2.

L203
Reference is necessary for FigTree v1.4, though I do not know how to cite a GitHub page.

L226,228,231
Consider omitting "(Figure 2)". I think it is enough citing at once in the top of this paragraph.

L236,238,239
Consider omitting "(Figure 3A)". I think it is better citing at once in the top of this paragraph.

L286
by comparing mtDNA and nDNA
How was this comparison in clupeids by Bloom & Lovejoy (2014)?

L287
such as it has been found in spined loach fish -> delete
Particular fish other than clupeids is not important. See above. Did this mitochondrial genomics deal with nDNAs?

L292
the sympatric species S. antipodum and S. muelleri from New Zealand (unclear) ->New Zealand S. antipodum and S. muelleri sympatric with each other

L298
Summarizing, the results in (verbose) -> delete

L306
that -> than ?

L315
2014)., -> 2014),

L322
Eoceno -> Eocene

L332
What does "lens" mean?

L334
a tropical submergence (uncountable) -> tropical submergence

L334-342
You should address an alternative parallel dispersal hypothesis of tropical MRCA to cool waters independently around 55.8 - c.a. 34 MYA.

L339,348
Grant, Leslie & Bowen -> Grant et al.

L366 reference
Reference list is still a den of errors. Check thoroughly again.

L375
Book title (not a series title)? Editors?

L386,394,etc
Paper titles should be in lower case.

L396
PloS one -> PloS One

L469
Journal titles should be in title case.

Fig.2 legend
L5-7
orange branches correspond to S. muelleri .. (verbose) -> orange to S. muelleri, light blue to S. antipodum, and green to S. sprattus

·

Basic reporting

English usage could be improved.

Experimental design

No comment.

Validity of the findings

No comment.

Additional comments

Thank you to the authors for their thoroughly revision of their manuscript. I don’t have major comment on the content and I think the results of this study are particularly interesting for the biogeography of Clupeoidei to understand their antitropical distribution.

However, I noticed several minor typos/misspelling/inaccuracies (see the list below) throughout the manuscript. Also, I am not an English native speaker, but the English usage could be edited.

Minor comments:
Line 45: sometimes the authors use “MYBP” and sometimes “MYA”. Usage needs to be homogenised throughout the manuscript.
Line 46: “in THE southern…”
Line 61: species is misspelled
Line 65: change “is” to “are”
Line 66: change “studies” to “studied”
Lines 69, 75 and 79: publication date after van der Laan is missing
Line 69: please revised “…are the important of…”
Line 170: change “suitable” to “informative”
Line 181: This sentence needs to be reworded: how many “consensus trees” did you obtain?
Line 187: “run” without “s”
Lines 191 and 197: MYA or MYBP?
Line 192: delete one “(Taverne 1971)”
Line 193: the darter in front of the fossil name Lecceclupea is missing
Line 194: “Ehirava, Gilchristella” in italics
Line 202: how many maximum clade credibility trees?
Line 226: change “is a valid species from a phylogenetic perspective” to “forms a monophyletic group”
Line 230/231: change “the species within the genus Sprattus are a paraphyletic group” to “the genus Sprattus is polyphyletic” because S. Sprattus is closely related to Clupea AND S. fuegensis, S. antipodum and S. muelleri are closely related to Ramnogaster and Strangomera.
Line 247 (and elsewhere as lines 257 and 258): I think polyphyletic is better than paraphyletic for the reason given above (under the comment on Line 230/231, above)
Line 259: “add” without “s”
Line 267/268: Linnaeus could not have done this because the genus Sprattus was described in 1846. Please reword.
Line 269: delete “on”
Line 288/289: this sentence needs to be reworded.
Line 294: who suggested? Need reference.
Line 322: “Eocene”
Line 434: “WAS” or “WS” as written in the above and below references

Likely more minor typos I missed. Please check carefully.

Figure 1 is nice but it could include the distribution of Ramnogaster and Strangomera as well.


A last comment: Potamalosa, Hyperlophus and Ethmidium are also temperate species, that are distributed in the southern hemisphere. Therefore, the “southern distribution” of the whole lineage predated by about 50 million years the origin of the clade (Sprattus, Clupea, Ramnogaster, Strangomera). This is well shown Figure 3 but not very visible in the text. As far as I know, this is one of the oldest south-distributed groups of fish. They were able to cross the tropics but not to adapt to warm environment.

---

## Round 0.3 · accepted · Accept

Thank you for addressing the reviewers' comments thoroughly.